# OpenReview forum: "Neural Frailty Machine: Beyond proportional hazard assumption in neural survival regressions"
_ICLR.cc/2023/Conference — Submitted to ICLR 2023_

### Official Review · Reviewer_oRcF · 2022-10-23

**Confidence:** 4
**Correctness:** 3
**Technical Novelty And Significance:** 3
**Empirical Novelty And Significance:** 2
**Recommendation:** 5

**Clarity, Quality, Novelty And Reproducibility:**

- The writing is good, the paper is easy to follow. There are some odd phrases, see below.
- The method is novel, however, not surprising as it seems mostly an extension of well-known classical survival techniques to the deep-learning setting and therefore relatively straightforward. The theoretical results seem non-trivial and strengthen the quality of the paper considerably.


Minor:

- Definition in equation (3) is the wrong way round, should be =: otherwise nonsensical.

> Regression analysis of time-to-event data (Kalbfleisch & Prentice, 2002) has been the most important modeling tool for clinical studies

"most important" is probably an exaggeration.

> The key feature that differentiates time-to-event data from other types of data is that they are incompletely observed, [...]

I would say "often" or "usually", but not necessarily.

**Strength And Weaknesses:**

__Strengths__:
- An interesting investigation into the area of frailty models from a neural network perspective
- The approach comes with interesting theoretical results concerning convergence rates.
- The simulation study covers the common survival literature, using standard datasets as well as comparing to many common approaches.

__Weaknesses__:
- Soden (Tang et. al. 2022) is a special case of earlier work of Groha et. al. (2020) which needs to be cited. Those authors also report better results on METABRIC than Soden or NFM here.
- The results in Table 1 have strongly overlapping confidence sets, casting doubts on the performance difference. Those might as well be explained by seed choice or hyperparameter tuning. Any claims on "superior predictive performance" should therefore be adjusted. (The term state-of-the-art should be fine, but explicitly superior seems too much.)

[1] https://arxiv.org/abs/2006.04893



**Summary Of The Paper:**

The paper introduces a new neural network technique for survival analysis termed the neural frailty model, which in turn is inspired by the literature on frailty models. The authors provide a theoretical analysis for the convergence rate of the conditional distributions of the event data and demonstrate state-of-the-art performance on common survival datasets.

**Summary Of The Review:**

The paper is strong in theory, but somewhat weak in the experiments section. In particular, claims are stronger than they should be and the current literature should be correctly presented. With these issues I don't think the paper is ready to be published.

---

> ### Author Response · Authors · 2022-11-11
> **Response to reviewer oRcF**
>
> We thank the reviewer for providing insightful comments. Below we address your specific points:
>
>
> **Q1: On the SurvNode model**
> Thanks for pointing out this work. We provide comparisons with SurvNode in the revision (see appendix D for details). SurvNode is yet another ode-based method for neural survival regressions. It was previously reported [1] and also empirically observed by us that ode-based methods are not scalable enough for large datasets. Given our limited computational resources, we are not able to evaluate the two ode-based methods, namely SurvNode and SODEN in large datasets like KKBOX.
>
>
> **Q2: Improvement in performance is marginal**
> The reason is mostly perhaps our evaluation protocol: Since five of six datasets we used do not have standard train/test splits and 5-fold cv is used for evaluation, we found unstable performance considering a single CV split. Therefore we perform 10 splits to obtain more stable performance. Similar evaluation protocol was considered in the SODEN paper [2]. As a consequence, most baselines are close in performance (this is also evident in the SODEN paper). Therefore, reporting best performance considering only mean metric is still reasonable for superiority, since otherwise all methods are identical, and no paper would be really meaningful.
>
>
> **Minor issues**
> We have fixed the minor issues in the revision according to your suggestions, thanks for pointing out.
>
>
>
> References:
> [1] Groha, Stefan, Sebastian M. Schmon, and Alexander Gusev. "A general framework for survival analysis and multi-state modelling." arXiv preprint arXiv:2006.04893 (2020).
> [2] Tang, Weijing, et al. "SODEN: A Scalable Continuous-Time Survival Model through Ordinary Differential Equation Networks." J. Mach. Learn. Res. 23 (2022): 34-1.

---

> > ### Comment · Reviewer_oRcF · 2022-11-16
> > **Still confused**
> >
> > __Q1__:
> > I am not sure Survnode is "another" ODE method. To me, SODEN looks like Survnode, but with the number of states set to 2. Since the method seems to appear earlier, the omission of the reference in many places where SODEN is referenced is odd. Similarly, the results of SODEN and Survnode should then be quite similar.
> >
> > __Q2__:
> > Indeed, the instability of the performance makes it difficult to rank different methods. The 10-fold split does show this difficulty in the width of the confidence bands.
> >
> > Yet, if I understand you correctly, there seem to be 2 arguments in your response in favour of ignoring those intervals:
> >
> > 1) _Performance evaluation as been done like this before in the SODEN paper._ This is not a good argument, as it would imply that any bad practice from a previous paper should be retained in future work.
> >
> > 2) _Under a rigorous interpretation of the experiment outcomes you cannot establish superiority of the method, therefore evidence thresholds need to be lowered._ This is similarly a bad argument. If an honest evaluation of the experiment lets us conclude that we cannot make a claim regarding superiority then this cannot be circumvented by lower standards because the authors assume superiority a necessity for meaningfulness.
> >
> > In my opinion, the method performs well and not worse than alternatives, but you cannot make the claim it's better. There simply isn't enough evidence. On this note, I don't think the difficulty to prove SOTA is a ground for rejection, however, unscientifically exaggerating the experimental outcomes is.

---

> > > ### Author Response · Authors · 2022-11-17
> > > **More on "honest" empirical comparisons**
> > >
> > > Firstly, we make a clarification here, **we did not use 10-fold cv, but use 10 independent cv splits, that constitutes 50 trial runs in total**. To get a clear understanding of instabilities, we list below the results on the metabric dataset using the NFM-PF model, grouped by 10 different cv splits. The results imply that cv performance differs across splits, by a margin (between the highest and lowest) of more than 1 percent. Therefore it is possible to report cherrypicked cv splits that exaggerates performance gain. We recognized this caveat of some previous studies and use a more stable evaluation method. We would like to know why you call it a **bad practice**.
> > > ||split 1|split 2|split 3|split 4|split 5|split 6|split 7|split 8|split 9|split 10|
> > > |---|---|---|---|---|---|---|---|---|---|---|
> > > |mean(cindex)|0.6516|0.6501|0.6456|0.6481|0.6477|0.6518|0.6517|0.6549|0.6504|0.647|
> > > |std(cindex)|0.0147|0.0215|0.0219|0.0201|0.0154|0.0089|0.0173|0.0118|0.0229|0.0241|
> > > |mean(ibs)|0.1628|0.1615|0.165|0.1651|0.1642|0.1593|0.1634|0.1653|0.1632|0.1641|
> > > |std(ibs)|0.0076|0.0067|0.0033|0.0097|0.0052|0.0057|0.0079|0.0097|0.0067|0.0078|
> > > |mean(inbll)|0.4903|0.4846|0.4945|0.4953|0.4924|0.4802|0.49|0.4949|0.4903|0.4946|
> > > |std(imbll)|0.023|0.0164|0.0097|0.0248|0.012|0.0142|0.0207|0.0233|0.0184|0.0217|
> > >
> > >
> > > Secondly, according to your comments, performance claims shall be reported on a statistically significant (i.e., using hypothesis testing techniques) scale. We agree that this provides stronger evidence than the evaluation protocol chosen in the paper. However, so far as we have noticed **there appears no previous works that used small scale datasets like METABRIC and RotGBSG (which exhibits large performance variance if cv is adopted), at the same time report performance claims with statistical significance** (Please kindly let us know if you have noticed such works). As a result, according to this stringent criterion, the situation in contemporary neural survival research is something like:
> > > - (i) Mean performance over cv splits are used for comparisons, variance scales are ignored due to lack of standard train/valid/test splits
> > > - (ii) No statistically dominant models (over prior models like RSF) have been discovered so far.
> > >
> > >
> > > If (i) is dishonest reporting, then according to our knowledge we have to face the frustrating situation of (ii). Therefore we respectfully disagree with your opinion that only statistically dominant claims shall be made.

---

> > > > ### Comment · Reviewer_oRcF · 2022-11-18
> > > > **Miscommunication?**
> > > >
> > > > Thank you for the clarifications regarding the CV splits.
> > > >
> > > > Perhaps some of my statements were not explained well. I will try to be more precise:
> > > >
> > > > > Secondly, according to your comments, performance claims shall be reported on a statistically significant
> > > >
> > > > Yes.
> > > >
> > > > > [...] there appears no previous works that used small scale datasets like METABRIC and RotGBSG [...] at the same time report performance claims with statistical significance
> > > >
> > > > That's what I mean with __bad practice__. The point is not to say that your studies are subpar when compared to previous studies. However, previous studies have pretty much all overstated the performance of their models, resulting in a situation where every proposed method is the best, while the overall performance on typical datasets like METABRIC hasn't really improved much.
> > > >
> > > > I appreciate that it is frustrating that suddenly more stringent assessment criteria are required, which is why I am okay with the paper being accepted even if superior performance cannot be demonstrated on the small dataset. However, the reporting standards currently employed in the field provide practitioners with very little evidence. We should not boldface and claim superior performance in instances where there exists high uncertainty and overlapping confidence bands.
> > > >
> > > > > Therefore we perform 10 different CV runs for each survival dataset and report average metrics as well as their standard deviations
> > > >
> > > > That's fine, but the confidence intervals would be overlapping, so your (correct) claim of the potential of cherry-picking applies even to the 10 independent splits.
> > > >
> > > > > If (i) is dishonest reporting, then according to our knowledge we have to face the frustrating situation of (ii). __Therefore__ we respectfully disagree with your opinion that only statistically dominant claims shall be made.
> > > >
> > > > I don't understand this argument. You are saying that because it's not possible to demonstrate superior performance, no such requirement should exist? That was _never_ the standard _anywhere_ in science. The request to take into account uncertainty should __not__ be contentious, it is a minimal scientific requirement in _any field_.
> > > >
> > > > By the way, the claim is not that "no statistically dominant models" have been found, but rather, that no improvement can be demonstrated on METABRIC / RotGBSG / FLCHAIN / SUPPORT.
> > > >
> > > > Your results on larger datasets like MIMIC and KKBOX seem to show a more significant difference. It would be great if you could carry out a significance test to confirm this.

---

### Official Review · Reviewer_nTQp · 2022-10-23

**Confidence:** 4
**Correctness:** 3
**Technical Novelty And Significance:** 3
**Empirical Novelty And Significance:** 2
**Recommendation:** 6

**Clarity, Quality, Novelty And Reproducibility:**

The presentation is clear. There is novelty but since it's a restricted version of Cox-Time (introduce more conditions on hazard), the novelty is not significant.

**Strength And Weaknesses:**

Strength:
1. The authors introduce deep networks in to frailty models.
2. The authors provide theoretical analysis and synthetic evaluations to verify the correctness of their approach.

Weakness:
1. I am still not very convinced that why we want to introduce a random variable to represent frailty. It reduces the representation power of the neural network based hazard model. Typically, the author uses a gamma family, why not use a more flexible family? In the real-world experiments, why do we need such a frailty model?I
2. The authors use a numerical integration to compute the integral in likelihood. What's the speed of this method compared to other approaches? Will it be very slow?
3. The authors should also include concordance as an evaluation metric in real-world study. It's a commonly used metric in survival analysis.
4. The authors may consider other recent SOTA methods, for example, survival mixture density networks (Han et al, 2022), DeSurv (Danks et al, 2022).


**Summary Of The Paper:**

The authors propose a neural network parameterized proportional frailty model. The proportional frailty model uses a random variable to express the unobserved heterogeneity corresponding to individuals. The main contribution is to introduce the neural networks into proportional frailty model.

**Summary Of The Review:**

I am not very convinced that why we need to introduce such a frailty model both makes the representation power smaller and computation time larger in a neural hazard model. More metrics and baselines should also be included. But I still think it's above the borderline since it's an interesting direction that includes some prior knowledge in deep models.

---

> ### Author Response · Authors · 2022-11-11
> **Response to reviewer nTQp**
>
> We thank the reviewer for providing insightful comments. Below we address your specific points:
>
>
> **Q1: The necessity of introducing frailty**
> We explain this in remark 1 in the paper and empirically verifies it in appendix D.5. In the PF scheme, introducing frailty significantly enhances the expressive power of the model (via going beyond proportional hazard assumption). It is commonly used in statistical literature for dealing with the unobserved hetrogeneity in data. In the FN scheme, although the frailty transform does not yield more expressive models in general (it does not reduce expressiveness), the inductive bias of frailty has proven to be useful in survival scenarios, see the comparison in appendix D.5 for more details.
>
>
> **Q2: Time complexity of numerical integration in the FN scheme**
> Performing numerical integration using quadrature methods is essentially a weighted sum over carefully chosen evaluation points (abscissas). In our experiments, we tested summation over 20, 50 and 100 points and found that summation over 20 points provides reasonable numerical convergence for evaluating the integral. Doing this weighted sum is very lightweight for modern GPU architectures since it can be completely parallelized. Even using only CPU,  we found NFMs to be very computationally efficient.
>
>
> **Q3: Report C-index**
> Performance comparison regarding the c-index metric is provided in appendix D.4 in our paper. The results suggest that on average, NFM is the best among all the chosen models.
>
>
> **Q4: More baseline comparisons: survival mixture density networks (SMDN), and DeSurv**
> We add DeSurv in our baseline comparisons in the revision. SMDN is contemporary work (posted on arXiv on August 2022) with ours.

---

### Official Review · Reviewer_yX4d · 2022-10-24

**Confidence:** 4
**Clarity, Quality, Novelty And Reproducibility:** See above.
**Correctness:** 3
**Technical Novelty And Significance:** 2
**Empirical Novelty And Significance:** 2
**Recommendation:** 5

**Strength And Weaknesses:**

**Strengths**
- The paper studies survival analysis, an important problem in many domains
- Modulating hazard functions with frailty models is interesting
- Theoretical convergence analysis of NFM-PF and NFM-FN optimising schemes may be of interest to some researchers
- The proposed approach seems easy to implement

**Weakness**
See references below.

*Technical Limitations*
- Its unclear why the gamma frailty model is advantageous compared to other frailty models apart from its popularity.
- The paper leverages Clenshaw-Curtis quadrature for numerical integration. It's unclear why other solvers are not considered, *e.g.*, adjoint [5].
- Its unclear how we should use the derived convergence rates for NFM-PF and NFM-FN schemes

*Underwhelming Experiments*
- Its unclear why the standard C-Index [7] is not used for model evaluation
- For completeness, the paper should also compare to models that directly estimate event times $p(t|x)$ parametrically [e.g., AFT, 1, 4] or non parametrically [2, 3].
-  Paper should provide qualitative results of estimated hazards $\lambda(t|Z)$, survival $S(t|Z)$ and  $p(t|Z)$ compared against ground truth times, particularly for the synthetic datasets
-  The synthetic experiment is not representative of real-world datasets:
(i) What happens when the frailty model is misspecified? Currently, the assumed frailty model matches the data generation mechanism;
(ii) The assumed very noisy hazard function seems unrealistic. Alternative data generation distributions described in [6] should be considered;
(iii) The performance gains in terms of IBS and INBLL seem marginal

**References**
- [1] Ranganath et al., "Deep survival analysis", In Machine Learning for Healthcare Conference, 2016
- [2] Chapfuwa et al., "Adversarial time-to-event modeling", ICML, 2018
- [3 Miscouridou et al., “Deep survival analysis: Nonparametrics and missingness,” Machine Learning for Healthcare Conference, 2018
- [4] Nagpal et al., "Deep Cox mixtures for survival regression", Machine Learning for Healthcare Conference, 2021
- [5] Chen et al.,  "Neural ordinary differential equation",  NeurIPS2018
- [6] Bender et al., "Generating survival times to simulate Cox proportional hazards models". Statistics in medicine (2005)
- [7] Harrell et al., “Regression modelling strategies for improved prognostic prediction,” Statistics in medicine, 1984

**Summary Of The Paper:**

The paper proposes gamma frailty for modulating proportional hazards (NFM-PF) or nonparametric hazards  (NFM-FN) functions. The assumed hazard functions  are modeled with neural networks where model parameters are learned via maximum likelihood.  Additionally, the paper provides  theoretical convergence analysis for the NFM-PF and NFM-FN optimising schemes. Experimental results on real-world datasets show improvements per metrics integral brier score  (IBS) and integrated negative binomial log-likelihood (INBLL).

**Summary Of The Review:**

Frailty survival models have been proposed before. Its unclear how we should use the derived theoretical convergence rates for NFM-PF and NFM-FN schemes. Additionally, the experimental results are unconvincing.

---

> ### Author Response · Authors · 2022-11-11
> **Response to reviewer yX4d**
>
> We thank the reviewer for providing insightful comments. Below we address your specific points:
>
>
> **Q1: Why is gamma frailty the preferred choice?**
> We introduce several other frequently used frailty configurations in appendix A of our paper, and in all the real-world data experiments we tune the choice of frailty distributions. Empirically, gamma frailty gives out the best performances in most cases.
>
>
> **Q2: Why not considering alternative methods for numerical integration?**
> Using quadrature type methods to numerically compute cumulative hazards has been a standard practice in survival analysis (for example see [1] for their procedure of estimating AFT). We refer the idea of using Clenshaw-Curtis quadrature to the work of UMNN paper[2], where the authors explained in detail the advantage of Clenshaw-Curtis quadrature over other types of quadrature methods like Newton-Cotes. The authors of [2] also reported that using less than 100 steps is sufficient for numerical convergence, in our experiments, we tested 20, 50 and 100 steps, and the result differences are neglegible. Since numerical convergence when using a relatively small number of integration steps is a desirable property, we do not think it is necessary to compare with other ODE methods.
>
>
> **Q3: Explain the usage of convergence rate**
> Establishing rates of convergence is a necessary step in introducing nonparametric regression models. As far as we know, the rates of convergence and related theories are not explored deeply for neural survival models. We provide a detailed and non-trivial derivation of the convergence rates of the proposed approach. The rates obtained in our paper are comparable to recent theoretical studies considering convergence properties of neural regressions. For the usage of the convergence results, generally speaking the rates obtained in our paper might not be minimax optimal, but the rates are still sufficiently fast for good performance------we empirically verify the result in section 5, stating that a thousand samples (which are smaller than all of the datasets used in the real-world study) are sufficient for convergence.
>
>
> **Q4: Report C-index**
> Performance comparison regarding the c-index metric is provided in appendix D.4 in our paper. The results suggest that on average, NFM is the best among all the chosen models.
>
>
> **Q5: More baseline comparisons**
> We add three more baseline methods in the revision.
>
>
> **Q6: Provide qualitative results of hazard estimates and survival estimates**
> We present pictorial illustrations of survival estimates compared with its ground truth values in appendix D.2 in the revision. Here we would like to make a further clarification of the illustration of synthetic experimental results in figure 1 of the paper: We plot $\hat{\nu}(T_i, Z_i)$ against $\nu(t_i, Z_i)$ for $i \in [100]$, and the horizontal axis stands for the index i, not the time points. The hazard curve itself is not noisy.
>
>
> **Q7: Synthetic experiments under misspecification of frailty**
> Misspecification is a very interesting and non-trivial question in frailty models [3]. Formally addressing this issue involves many more theoretical tools and principled experimental comparisons, and is therefore beyond the scope of our paper. Nonetheless, we make some preliminary investigations of robustness properties of NFM: Using the synthetic datasets, we use gamma frailty as the true model and Box-Cox frailty (see appendix A for detailed formulations) as the frailty distribution used for estimation, the resulting survival curves are plotted with sample size 1000 and 5000 in [this url](https://imgur.com/a/bMPzp0T) . The results implies robustness properties between gamma and Box-Cox frailties. Precise characterizations of relationships between frailty distributions is left for further explorations.
>
> **Q8: The performance gains in terms of IBS and INBLL seem marginal**.
>  Please refer to our response to reviewer oRcF for a detailed explanation. The reason is mainly because of the evaluation protocol.
>
> References:
> [1] Ding, Ying, and Bin Nan. "A sieve M-theorem for bundled parameters in semiparametric models, with application to the efficient estimation in a linear model for censored data." The Annals of Statistics 39.6 (2011): 3032-3061.
> [2] Wehenkel, Antoine, and Gilles Louppe. "Unconstrained monotonic neural networks." Advances in neural information processing systems 32 (2019).
> [3] Kosorok, Michael R., Bee Leng Lee, and Jason P. Fine. "Robust inference for univariate proportional hazards frailty regression models." The Annals of Statistics 32.4 (2004): 1448-1491.

---

### Official Review · Reviewer_xG5j · 2022-10-31

**Confidence:** 3
**Correctness:** 4
**Technical Novelty And Significance:** 3
**Empirical Novelty And Significance:** 3
**Recommendation:** 6

**Clarity, Quality, Novelty And Reproducibility:**

The paper is very well-written and the authors have made their contributions clear. The work is novel and reproducible.

**Strength And Weaknesses:**

I found this paper interesting. It is well-written and the theoretical results seem interesting. However, section 4 is quite disconnected. The authors seem to get tangled up into explaining their theory and in my eyes lose sight of the paper flow. I would have appreciated a better narrative of what is the significance of their results. In other words, what do we gain from knowing the rate of convergence? Does it show that the proposed method is efficient? Or on the contrary it shows that it is slow?

Given the survival analysis literature has paid considerable attention to time-varying covariates recently, a drawback of this work is failing to consider time-varying covariates.

**Summary Of The Paper:**

This work proposes a hazard estimator which models the unobserved heterogeneity in individuals using frailty models. They propose two function approximation schemes where in one they assume the covariate effect and time effect to be multiplicative and in the other no assumptions are made. They also provide theoretical results on the convergence rate of their estimators.

**Summary Of The Review:**

I find this work novel and interesting. Nonetheless, it only handles time-static covariates which is a major drawback given the recent advances in the time-varying survival analysis literature.

---

> ### Author Response · Authors · 2022-11-11
> **Response to reviewer xG5j**
>
> We thank the reviewer for providing insightful comments. Below we address your specific points:
>
>
> **Q1: Provide a better explanation of consequences of convergence rate**
> Establishing rates of convergence is a necessary step in introducing nonparametric regression models. So far as we have noticed, such statements are rarely seen in neural survival proposals, and we provide a detailed and non-trivial proof, as well as a standard type of technique to discuss the correctness of corresponding methods. The rates obtained in the paper are comparable to recent theoretical studies considering convergence properties of neural regressions[1]. A very nice question is the efficiency of our theoretical results------we understand this as whether the rates obtained are optimal in the minimax sense. Several works claim that their rates are optimal in the minimax sense up to logarithmic factors, but requires the true underlying function to lie in a smaller function space (i.e., the formualtion of [2]). Generally speaking, the rates obtained in our paper might not be minimax optimal (similar statements was also present in [1]), but the rates are still sufficiently fast for good performance------we empirically verify the result in section 5, stating that a thousand samples (which are smaller than all of the datasets used in the real-world study) are sufficient for convergence.
>
>
> **Q2: Ability to consider time-varying covariates**
> Time-varying covariates is an interesting topic. From a theoretical standpoint, incorporating time-varying convariates into the modeling mechanism is trivial under the counting process formulation [3]. However, from an empirical viewpoint, datasets with time-varying covariates are mostly small in scale (such as the datasets provided in [4]), making nonparametric/neural modeling less stable. Therefore, we do not consider generalizing to time-dependent covariates in our paper.
>
>
> References:
> [1] Farrell, Max H., Tengyuan Liang, and Sanjog Misra. "Deep neural networks for estimation and inference." Econometrica 89.1 (2021): 181-213.
> [2] Schmidt-Hieber, Johannes. "Nonparametric regression using deep neural networks with ReLU activation function." The Annals of Statistics 48.4 (2020): 1875-1897.
> [3] Zeng, Donglin, and D. Y. Lin. "Efficient estimation of semiparametric transformation models for counting processes." Biometrika 93.3 (2006): 627-640.
> [4] Rondeau, Virginie, Yassin Marzroui, and Juan R. Gonzalez. "frailtypack: an R package for the analysis of correlated survival data with frailty models using penalized likelihood estimation or parametrical estimation." Journal of Statistical Software 47 (2012): 1-28.

---

### Author Response · Authors · 2022-11-11
**Revision**

We'd like to thank all the reviewers for providing helpful comments. We've made the following updates to our paper based on your feedback:
- In the synthetic experiments, we provide additional comparisons of survival predictions against the true survival curve in appendix section D.2.
- We add three state-of-the-art models: SurvNode[1], DCM[2] and DeSurv[3] in all our baseline comparisons. And according to our experimental evaluations, NFM exhibits better performance on 5 of the 6 benchmark datasets than all the baseline models

Due to page limits, we place most of the revised contents in appendix D. Further, we respond to individual comments below.

References:
[1] Groha, Stefan, Sebastian M. Schmon, and Alexander Gusev. "A general framework for survival analysis and multi-state modelling." arXiv preprint arXiv:2006.04893 (2020).
[2] Nagpal, Chirag, et al. "Deep Cox mixtures for survival regression." Machine Learning for Healthcare Conference. PMLR, 2021.
[3] Danks, Dominic, and Christopher Yau. "Derivative-Based Neural Modelling of Cumulative Distribution Functions for Survival Analysis."   International Conference on Artificial Intelligence and Statistics. PMLR, 2022.

---

### Decision · Program_Chairs · 2023-01-20

**Decision:**

Reject

**Justification For Why Not Higher Score:**

A higher score is possible if the authors have provided clearer motivation and reasoning on when the proposed NFM is expected to be better than existing methods, and included convincing ablation study on various modeling choices.

**Justification For Why Not Lower Score:**

N/A

**Metareview: Summary, Strengths And Weaknesses:**

The paper proposes Neural Frailty Machines (NFM) that model the hazard function in survival analysis with the product of a gamma random variable $\omega$, which models the frailty distribution, and a neural-network-based non-negative function of the time $t$ and covariates $Z$. It optimizes NFM with an observed log-likelihood (OLL) objective, and presents theoretical properties on rates of convergence assuming the underlying event data follows the assumption of NFM. Empirical results on synthetic data and real-world data are used to verify the performance of NFM.

Several concerns on this paper make the AC hesitant to recommend for acceptance:

1. There exist previous works on using neural network for survival analysis, including the ones pointed out by Reviewer yX4d that directly model the event times using neural network-powered distributions. Discussing the advantages of NFM over neural survival models that directly model event times, and adding performance comparison would help clearly strengthen the paper.

2. Some modeling choices, including why using gamma frailty distribution, are not clearly motivated and justified. In particular, under the current construction, NFM assumes the hazard rate follows a gamma distribution, the log of whose scale/rate is linked to the time and covariates via a neural network. It is unclear why the parametric gamma assumption on the hazard function is not a restrictive assumption, in particular, in comparison to existing neural survival models.

3. Lack of ablation study on various modeling components, and lack of explanations on when and where they are expected to boost the performance.

4. Theoretical analysis in Section 4 feels disconnected from the other parts of the paper. While it justifies the proposed estimates when the NFM assumption is correct, it provides little insight on how the algorithms would behave when the NFM assumption is violated.

5. The experimental results are not strong enough to convince the reviewers to trust NFM as a better choice over existing methods.

The paper would benefit from a careful revision to address these concerns.

**Summary Of Ac-Reviewer Meeting:**

Two reviewers participated in a meeting with the AC. We discussed both the merits of the paper and what parts remain unconvincing given the authors' rebuttal. The remaining concerns of the reviewers have been summarized into the metareview.